# Supplemental Oxygen in the Newborn: Historical Perspective and Current Trends

**DOI:** 10.3390/antiox10121879

**Published:** 2021-11-25

**Authors:** Maxwell Mathias, Jill Chang, Marta Perez, Ola Saugstad

**Affiliations:** 1Center for Pregnancy and Newborn Research, Department of Pediatrics, Section of Neonatal-Perinatal Medicine, University of Oklahoma Health Sciences Center, Oklahoma City, OK 73104, USA; 2Division of Neonatology, Department of Pediatrics, Northwestern University Feinberg School of Medicine, Chicago, IL 60611, USA; jchang@luriechildrens.org (J.C.); mtperez@luriechildrens.org (M.P.); o.d.saugstad@medisin.uio.no (O.S.); 3Ann & Robert H. Lurie Children’s Hospital of Chicago, Chicago, IL 60611, USA; 4Department of Pediatric Research, University of Oslo, N-0424 Oslo, Norway

**Keywords:** hyperoxia, prematurity, bronchopulmonary dysplasia, retinopathy of prematurity

## Abstract

Oxygen is the final electron acceptor in aerobic respiration, and a lack of oxygen can result in bioenergetic failure and cell death. Thus, administration of supplemental concentrations of oxygen to overcome barriers to tissue oxygen delivery (e.g., heart failure, lung disease, ischemia), can rescue dying cells where cellular oxygen content is low. However, the balance of oxygen delivery and oxygen consumption relies on tightly controlled oxygen gradients and compartmentalized redox potential. While therapeutic oxygen delivery can be life-saving, it can disrupt growth and development, impair bioenergetic function, and induce inflammation. Newborns, and premature newborns especially, have features that confer particular susceptibility to hyperoxic injury due to oxidative stress. In this review, we will describe the unique features of newborn redox physiology and antioxidant defenses, the history of therapeutic oxygen use in this population and its role in disease, and clinical trends in the use of therapeutic oxygen and mitigation of neonatal oxidative injury.

## 1. Introduction

### 1.1. Hyperoxia Is Damaging to Developing Organ Systems

Mammals gestate at lower oxygen tension than their postnatal environment, and birth (even in the absence of supplemental oxygen) constitutes an increase in oxygen exposure [1]. In addition, rapid post-natal growth and development requires significant oxygen delivery and consumption per gram of tissue in newborns, and energy sources are more rapidly depleted than in adults [2,3]. Alterations in mitochondrial oxygen concentration can induce apoptotic cell signaling pathways through mitochondrial proton leak [4], while non-mitochondrial derived reactive oxygen and nitrogen intermediates (ROI) play an important role in intra- and inter-cellular growth factor signaling (see [5] for review). Lastly, enzymatic and non-enzymatic antioxidant systems in newborns have decreased capacity to sequester ROI [6]. Premature exposure to ambient air (fraction of inspired oxygen or FiO_2_ = 21%) and oxygen exposure above ambient air (FiO_2_ > 21%) disrupts these processes.

The preterm and term newborn occupies a unique position as an organism with high tissue oxygen demand, growing and developing organ systems, and susceptibility to oxidative injury. Excess oxygen is the substrate for free radical formation by several enzymes, such as the xanthine oxidase and NADPH-oxidase families [7]. Free radicals induce membrane disruption and activation of inflammatory pathways through lipid peroxidation, affecting multiple organ systems. Elevated lipid peroxidation products have been found in plasma samples of infants with bronchopulmonary dysplasia (BPD) compared to those without [8]. In the lungs, hyperoxia exposure induces alveolar simplification and vascular remodeling, which is associated with disruption in the electron transport chain [9,10]. Hyperoxia exposure is also associated with injury and altered physiology in the developing intestines and retinopathy of prematurity (ROP), a developmental eye disease involving aberrant growth of the retinal vasculature that can result in blindness [11,12,13]. Lastly, neonatal hyperoxia alters cerebral blood flow, and neuronal apoptosis and inflammation have been shown after neonatal hyperoxia exposure [14,15,16].

### 1.2. Full Term and Preterm Newborns Have Decreased Antioxidant Capacity

Many important vitamin and mineral stores are delivered to the fetus during the third trimester of pregnancy; as a result, preterm infants have decreased levels of these important cofactors relative to term infants [17]. Plasma vitamin C levels are lower in preterm infants relative to term infants [18]. In a prospective study of vitamin D supplementation in 100 preterm infants, two-thirds had biochemical vitamin D deficiency [19]. In addition, pulmonary levels of glutathione peroxidase 1, cytosolic superoxide dismutase (SOD), mitochondrial SOD, and extracellular SOD are significantly lower in newborn mice than those at 3 and 7 weeks of age in mice. Along with decreased antioxidant expression, newborn mice exhibit increased mitochondrial oxidative stress compared to adults during hyperoxia exposure [6]. In a cohort of preterm infants with and without BPD, plasma vitamin E levels were below term infant levels, and were lower in infants with BPD than those without [20].

Given the decreased levels of antioxidant enzymes and vitamins in preterm infants, several clinical trials have been conducted to evaluate whether supplementation with either intact antioxidant enzyme or vitamin cofactors might decrease morbidity and mortality in this population. Trials of administration of exogenous SOD in the clinical setting were initially promising [21,22,23]; however, meta-analyses and long term follow-ups have shown no benefit over placebo [24]. Similarly, supplementation with the antioxidant cofactor vitamins A, C, and E have been studied, along with the glutathione precursor N-acetylcysteine (NAC). Vitamin A has been the subject of extensive research with mixed results, including the largest randomized controlled trial to date entitled NeoVitaA, which has not yet published its results [25,26,27]. A Cochrane meta-analysis in 2016 found marginal benefit of intramuscular administration of vitamin A to prevent death or BPD (RR 0.93 [0.88, 0.99]); however, intramuscular formulation is not readily available and repeat intramuscular injections pose a practical challenge in very low birth weight (<1500 g) infants [28]. Vitamin C, E, and NAC have also not shown clear benefit, and supplementation beyond nutritional needs is not recommended [29,30,31,32,33,34].

### 1.3. ROI Are Essential Components of Cell Signaling Pathways in Development

Despite the association of neonatal disease with ROI byproducts (e.g., lipid peroxides), the growth and development of organs and tissues relies on the production of ROI. Numerous growth factor receptors are tyrosine kinases (RTKs), whose function depends on local hydrogen peroxide derived from ROI. Peroxide oxidizes protein phosphatases (PTPs) by altering the 3-D structure and allowing RTK autophosphorylation and signal transduction [35,36]. Suppression of hydrogen peroxide with exogenous catalase will reduce tyrosine kinase phosphorylation and decrease growth factor signaling, which has been shown to reduce vascular endothelial cell migration [37,38]. In addition, increases in antioxidant capacity can impair growth and development. Inhibition of nuclear factor erythroid 2-related factor 2 (NRF2), an important transcription factor in the production of antioxidants in response to oxidative stress, increased fetal growth in a mouse model of pregnancy-associated hypertension, suggesting an important role for ROI in fetal growth [39]. A clearer picture of the complex interplay between postnatal oxygen exposure, antioxidant expression and function, and development and growth signaling is needed.

## 2. Supplemental Oxygen in the Initial Resuscitation of Newborns

### 2.1. Historical Aspects

While the first documented use of oxygen in infants was in 1780, just few years after its discovery, it was not until 1928 that a description of using oxygen in newborn resuscitation was published [40]. Oxygen therapy was subsequently introduced into routine newborn care in the 1930s and early 1940s. This was followed in relatively short order by the first published description of retrolental fibroplasia (RLF), a mysterious eye disease found in a small group of infants born prematurely, now known as ROP [41]. The progressive nature of this disorder was confirmed in 1948 by a husband-and-wife opthalmologist team working at Johns Hopkins University, who described development of RLF changes in a cohort of premature infants [42]. It was not until the 1950s that oxygen was identified as the culprit behind RLF, the leading cause of blindness in children of preschool age at that time [43,44,45]. This discovery was followed by decades of avoidance of oxygen that likely led to decreased survival of the most premature and sickest infants [46]. With the advent of oxygen saturation monitoring in the 1980s, however, oxygen therapy could be better targeted, resulting in improved survival and an associated increase in ROP once again [47].

### 2.2. Supplemental Oxygen in Term Infants

By the 1960s, the use of 100% oxygen in the delivery room was viewed as the standard of care and the most sensible approach to resuscitation of asphyxiated infants [48]. The International Liaison Committee on Resuscitation (ILCOR), first formed in 1992 to provide a forum for major resuscitation organizations in the industrialized world, recommended the use of 100% oxygen in its first set of newborn guidelines [49]. There were published opinions against such an approach as early as 1980, based on emerging clinical and animal data that implicated delivery room hyperoxia in subsequent oxidative stress generation [7,50]. This included evidence of increased hypoxanthine concentrations in the cerebral cortex of hypoxemic piglets randomized to resuscitation with 100% FiO_2_ compared to 21% FiO_2_, suggesting the presence of more severe energy metabolism deficits in the hyperoxia-exposed group. Animal models also increasingly supported the use of 21% FiO_2_ during resuscitation and demonstrated that it is as effective as 100% FiO_2_ in the resuscitation of hypoxemic piglets, with similar improvements in vital signs, base deficit, and plasma hypoxanthine [51,52]. Furthermore, studies in human neonates demonstrated that term newborns can be adequately resuscitated with 21% FiO_2_, with lower mortality in infants exposed to 21% vs. 100% FiO_2_ [53], with follow-up studies showing similar long-term neurodevelopmental outcomes [54]. Additional studies demonstrated that resuscitation with 100% FiO_2_ can be harmful [55,56]. In term neonates with perinatal asphyxia, resuscitation with 21% FiO_2_ resulted in more rapid improvements in Agpar scores and faster onset of spontaneous respiration, while infants treated with high FiO_2_ demonstrated evidence of increased oxidative stress at one month of age, with lower reduced-to-oxidized glutathione ratios and higher erythrocyte antioxidant enzyme activity [55]. Finally, a meta-analysis also contributed to the growing consensus; in term and late-preterm neonates in need of resuscitation at birth, 21% FiO_2_ restores heart rate and spontaneous respiration as effectively as 100% FiO_2_, while also being associated with significant reductions in neonatal mortality [57]. This accumulating evidence resulted in revisions to neonatal resuscitation guidelines that now recommend starting with 21% FiO_2_ and caution against 100% FiO_2_ in near-term and in-term infants [58,59].

### 2.3. Supplemental Oxygen in Preterm Infants

Evidence for the optimal oxygen concentration in the resuscitation of preterm infants is less conclusive. The available data suggest that many preterm infants need an FiO_2_ somewhere between 21% and 100% during resuscitation. A recently published meta-analysis found conflicting data on the impact of initiating resuscitation with lower (<40%) vs. higher (≥40%) FiO_2_ in preterm infants, with 9 out of 10 studies finding no differences in mortality [60]. Other publications, however, found an association between initiation of resuscitation with lower FiO_2_ and increased risk of death in immature preterm infants in an international cohort [61] and a Canadian cohort [62]. However, a follow-up study of the Canadian cohort at 18 to 21 months corrected age (gestational age at birth plus absolute age) found no differences in the composite outcome of death or neurodevelopmental impairment and an increased risk of severe neurodevelopmental impairment in infants who received 100% FiO_2_ versus those receiving 21% FiO_2_ [63]. However, preterm infants not reaching an oxygen saturation (SpO_2_) of 80% within the first 5 min of life have been shown to have higher mortality and risk of severe IVH [64,65]. In addition, the combination of bradycardia and hypoxemia in the first minutes of life wasn’t increases mortality. Although the optimal initial FiO_2_ is not fully known in these infants, it is still recommended to target a SpO_2_ of 80–85% within the first 5 min of life, as well as to avoid bradycardia (heart rate < 100 bpm).

## 3. Supplemental Oxygen in the Neonatal ICU

### 3.1. Historical Perspective

Beyond the delivery room, exposure to hyperoxia and oxidative stress has been associated with a multitude of adverse outcomes in infants, including ROP, BPD, and brain injury (including intraventricular hemorrhage and periventricular leukomalacia). These concerns have led to improvements in methods for measuring oxygenation in neonates, enabling a more precise titration of oxygen delivery. However, the optimal oxygen saturation target in extremely premature infants remains uncertain given the varying results in both randomized and observational studies [66]. Trials conducted in the 1950s demonstrated that administration of high FiO_2_ to preterm infants significantly increased their risk of severe ROP and blindness and led to the practice of restricting FiO_2_ in the 1960s to no more than 50% [45,67,68,69,70]. This change was estimated to result in an excess of 16 deaths per case of blindness prevented [46]. The introduction of transcutaneous PO_2_ electrodes allowed for more precise and tighter control of oxygen delivery to preterm infants, and a reduction in ROP was seen in all but the lowest gestational ages. In the 1980s and 1990s, pulse oximetry became the preferred method for monitoring oxygenation, while in the early 2000s, the American Academy of Pediatrics (AAP) suggested a target SpO_2_ range of 85–95%, which corresponds to PaO_2_ range of 29–67mm Hg (3.8–8.9 kPa), in oxygen-dependent preterm babies in the first 2 weeks of life [71].

### 3.2. Targeted Use of Supplemental Oxygen

In 2003, the Neonatal Oxygenation Prospective Meta-Analysis (NeOProM) collaborative study was formed to address the question of ideal oxygen targets in extremely preterm infants. The goal was to examine the effects of low versus high functional oxygen saturation targets in the postnatal period in premature infants <28 weeks gestation. Investigators in separate randomized clinical trials prospectively planned to undertake individual trials using similar study designs, participants, interventions, comparators, and outcomes. They agreed to provide individual participant data at trial completion for inclusion in a meta-analysis [69]. Prior to this collaborative effort, there existed only a few small randomized trials in the 1950s [44,72,73,74] and in the early 2000s [75] investigating oxygen targets in preterm infants. NeOProM ultimately included five multicenter studies under five study groups: Surfactant, Positive Pressure, and Pulse Oximetry Randomized Trial (SUPPORT) in the United States [76]; Benefits of Oxygen Saturation Targeting (BOOST II) trials conducted separately in the United Kingdom, Australia, and New Zealand [77,78,79]; and the Canadian Oxygen Trial (COT) [80]. In these 5 trials, a total of 4965 premature infants (<28 weeks gestation) were randomized to either a low (85–89%) or high (91–95%) SpO_2_ within the first 24 h after birth. Subsequent meta-analyses including follow-up data from the NEOPROM studies have since been published [69,81,82,83].

The primary outcome for each of the trials was a composite of death or disability by 18–24 months corrected age. No differences in the primary outcome were found between infants assigned to the low (85–89%) or high (91–95%) SpO_2_ range. Sub-group analysis also showed no differences for the primary outcome (death or major disability) for gestational age, inborn vs. outborn status, antenatal corticosteroid use, sex, small for gestational age (SGA) vs. appropriate for gestational age (AGA), singleton vs. multiple birth, type of delivery, age at the start of the intervention, or oximeter software type [84].

Despite finding no difference in primary outcome, secondary outcomes from the NeOProM trial raised concerns regarding the importance of oxygen exposure. A higher rate of mortality before the corrected age of 36 weeks, before hospital discharge, and prior to reaching a corrected age of 18 to 24 months was found in infants assigned to the low SpO_2_ (85–89%) target range [69,85]. A higher incidence of necrotizing enterocolitis (NEC) was also associated with the low SpO_2_ (85–89%) target range. In contrast, there was a lower incidence of ROP requiring treatment associated with the low SpO_2_ (85–89%) target range [69,81,82,83,84]. Of note, this finding showed substantial heterogeneity because SUPPORT was the only trial within NeOProM to find significant reductions in the need for ROP treatment [85]. Additionally, the increased incidence of ROP in the higher oxygen group (91–95%) did not translate to increased severe visual impairment (defined as bilateral legal blindness) at 18 to 24 months [83]. Additional secondary outcomes including physiologic BPD, intraventricular hemorrhage (IVH), periventricular leukomalacia (PVL), and neurodevelopmental outcomes as assessed by using the Gross Motor Function Classification System or Bayley Scales of Infant Development (BSID) did not differ between the low- and high-oxygen groups [81,82,86].

One main limitation of the NeoProM trial was that significant overlap was found in the actual exposed oxygen saturation between the two groups, despite the design of the trials specifying separation between the target ranges. Although the protocols specified a distinct separation of the SpO_2_ target in the two groups there was significant overlap in the SpO_2_ achieved, resulting in poor separation between the intervention and comparison groups [82]. Poststudy analysis showed that subjects randomized to 91–95% spent 13.9–22.4% of the time with SpO_2_ >95% and subjects randomized to 85–89% spent 20.2–27.4% of the time <85% while on supplemental oxygen [83]. An additional limitation to the NeOProM trial was that two trials (BOOST II trials in the United Kingdom and Australia) were stopped early, which may have resulted in some overestimation of the effect on mortality in these trials [84].

Following the NeOProM publications, some experts in the field suggested that functional SpO_2_ should be targeted at 90–95% in infants with gestational age <28 weeks until 36 weeks corrected age and alarm limits 89–95% [71,81]. In 2016, the AAP released new guidelines stating that “the ideal physiologic target range for oxygen saturation for infants of extremely low birth weight is likely patient-specific, dynamic, and dependent on various factors, including gestational age, chronological age, underlying disease, and transfusion status” [86]. Additionally, these guidelines conclude that a target oxygen saturation “range of 90% to 95% may be safer than 85% to 89%, at least for some infants.” The 2019 European guidelines recommend “in preterm babies receiving oxygen, the saturation target should be between 90 and 94%” and that “alarm limits should be set to 89 and 95%” [87].

Both the SUPPORT and BOOST trials used fixed, binary SpO_2_ targets for the duration of study enrollment (birth to 36 weeks corrected age). It is possible that optimal oxygen saturation varies during the course of NICU hospitalization. The pathophysiology of ROP is thought to involve initial hyperoxic arrest of vascular growth followed by hypoxia-induced hyperproliferation of the retinal vasculature [13,88,89]. In the STOP-ROP trial, infants were enrolled at the time of diagnosis of ROP (mean 10 weeks of absolute age or 35 weeks corrected age) to receive oxygen to target SpO_2_ to 89–94% or 96–99%. While the higher oxygen saturation target was associated with adverse pulmonary events and prolonged hospital stay, it was not associated with progression of ROP [75]. In a retrospective study comparing a static SpO_2_ goal to a biphasic goal that increased the target to >95% at 34 weeks corrected, infants in the biphasic group had a lower incidence and severity of ROP without an impact on mortality [90].

### 3.3. Persistent Pulmonary Hypertension of the Newborn

One notable exception to the targeting of specific SpO_2_ in the newborn is for infants with persistent pulmonary hypertension of the newborn (PPHN). Prenatally, most of the fetal blood flow bypasses the lungs through elevated pulmonary blood pressure and right-to-left shunts at the level of the atria through the foramen ovale and the great arteries through the ductus arteriosus. During normal birth, crying and initiation of the first breaths results in a rapid decrease in pulmonary blood pressure, allowing blood to circulate through the lungs and participate in gas exchange to oxygenate the infant. When infants do not undergo this physiologic decrease in pulmonary blood pressure, they develop PPHN.

The severity of PPHN can range from mild, requiring only brief oxygen supplementation, to severe, requiring intubation, mechanical ventilation, and even cardiopulmonary bypass with extracorporeal membranous oxygenation (ECMO) [91]. Oxygen is a potent pulmonary vasodilator within the context of hypoxia [92]; as such, up to 100% FiO_2_ is commonly used in the treatment of PPHN, regardless of infant SpO_2_ [93]. However, there is no clear benefit and potential harm in using such high FiO_2_ in the treatment of PPHN. In a lamb model of meconium-aspiration-induced PPHN, pulmonary blood flow and brain oxygen delivery were maximized when targeting a SpO_2_ of 95–99% during a 6 h exposure when compared to a fixed FiO_2_ of 100% or targeting SpO_2_ of 85–89% or 90–94%. Notably, the 95–99% target group showed a mean FiO_2_ of 50% and significantly higher pulmonary blood flow than the 100% FiO_2_ group [94]. In addition, it is likely that the vasodilatory effect of hyperoxia is short-lived. In fetal lambs exposed to 100% FiO_2_, hyperoxic pulmonary vasodilation peaked at approximately 50 min and decreased therafter [95]. Lastly, alveolar hyperoxia may increase pulmonary vascular contractility and impair responsiveness to pulmonary vasodilators, most notably inhaled nitric oxide, worsening outcomes in PPHN [96,97]. The optimal SpO_2_ in infants with PPHN remains unclear and requires further study.

## 4. Future Directions

### 4.1. Limits of Pulse Oximetry

All of the above studies rely on peripheral pulse oximetry measurements to guide supplemental oxygen administration. Refinements to oximeters and their software have improved portability and reliability; however, they continue to provide only a single data point (arterial oxygen saturation) to drive clinical decision making. The above studies were designed to optimize tissue oxygen delivery without exposing infants to excess oxygen; however, optimal oxgyen delivery depends on other factors, including hemoglobin concentration, tissue oxygen consumption, and hemodynamic status, among a multitude of others [98,99].

In addition, preterm infants have predominantly fetal hemoglobin (HbF), which binds more avidly to oxygen than the predominant adult form of hemoglobin (HbA) [100]. The difference in relative oxygen affinity affects oxyhemoglobin dissociation and delivery to the tissues; thus, an infant with 95% oxyhemoglobin and predominantly HbF may deliver less oxygen to the tissues than an infant with 90% oxyhemoglobin and predominantly HbA. While the ratio of HbF-to-HbA is highly associated with gestational age, interventions to improve oxygen delivery, such as blood transfusions (which are by necessity adult blood), can have a large impact on the amounts of each [101].

### 4.2. Near-Infrared Spetroscopy

One technology that provides data for other aspects of tissue oxygen delivery is near-infrared spectroscopy (NIRS). As with pulse oximetry, NIRS calculates the ratio of oxyhemoglobin to deoxyhemoglobin, but it does not identify the pulse waveform; therefore, it includes non-arterial sources of transmitted light from the monitored region [102,103]. Because they are measurements of regional oxygen saturation, NIRS measurements are affected by both oxygen consumption and oxygen delivery. The relative contributions of arterial (oxygen delivery) and venous (oxygen consumption) oxygen saturations are 30% and 70%, respectively [104,105]. Cerebral regional oxygen saturation (CrSO_2_) values have been found to be closely correlated with central venous oxygen saturation, a measure commonly used to calculate global oxygen consumption [106]. There are no established standards for use in neonates; however, multiple prior and ongoing research projects have investigated its use in monitoring regional oxygen saturation, most commonly in the brain and kidneys, and have proposed possible uses for these data [106,107,108,109,110].

A pilot study of very preterm (<32 week) infants using the combination of peripheral pulse oximetry and CrSO_2_ found significant differences in combined values between infants who developed IVH between 12 and 72 h of life and those who did not [111]. Another study looking specifically at the newborn transition period (first 15 min of life) in very preterm infants found significantly lower CrSO_2_ in infants who went on to develop IVH in the first 14 days of life [112]. A European unblinded, randomized, controlled trial using CrSO_2_ to guide therapy, entitled Safeguarding the Brain of Our Smallest Children (SafeBoosC), set a target range for CrSO_2_ of 55% to 85% and provided 8 treatment options when CrSO_2_ went below 55%. Notably, only one option involved increasing FiO_2_ [113]. The trial found a decrease in cerebral hypoxia (CrSO_2_ < 55%) but no differences in rates of brain injury or neurodevelopmental outcomes at 2 years of age [114]. A systematic review of CrSO_2_-guided therapy trials found only one trial that met the quality parameters set by reviewers and did not show a reduction in brain injury with the use of cerebral NIRS monitoring [115].

Renal regional oxygen saturation (RrSO_2_) may also provide enhanced data to guide therapy. It is possible that RrSO_2_ changes more rapidly than CrSO_2_ in response to discrepant oxygen delivery and oxygen consumption, as cerebral autoregulation protects the brain against rapid changes in perfusion in response to changes in hemodynamics [116]. Interestingly, higher RrSO_2_ in neonates with hypoxic ischemic encephalopathy was shown to be associated with acute kidney injury [117]. In contrast, in infants undergoing heart surgery, low RrSO_2_ was associated with the development of acute kidney injury [118].

Specific guidelines for the use and interpretation of NIRS may provide a framework to apply this technology to specific clinical scenarios to optimize oxygen delivery in the NICU [119]. The simplicity of pulse oximetry makes it an ideal monitor to guide clinical therapy, and the refinement of oxygen therapy guidelines using pulse oximetry has clearly shown benefit. The question remains whether the precision afforded by the addition of NIRS monitoring has enough benefit to overcome the burden of additional monitors on fragile infants in the NICU with limited body surface area.

### 4.3. Novel Therapies

In addition to limitations in monitoring, there are no available therapies to mitigate the effects of oxidative injury in the NICU. In a review of antioxidant therapies with particular focus on BPD, Ofman and Tipple outlined major barriers to effective therapies, including lack of compartment and target specificity, limited bioavailability, timing of therapy, and genetic variability [120]. Endogenous antioxidant enzymes are highly localized and perform specific functions in redox biochemistry. Systemic or even organ-specific administration of antioxidants may not reach the cells or organelles where hyperoxia induces injury. In addition, individual organs have variable oxygen consumption rates and respond differently to hypoxia and hyperoxia. This is evident in the NeOProM trials, where lower SpO_2_ targets were associated with NEC and mortality, while higher SpO_2_ targets were associated with ROP. Future therapies could target oxygen delivery specifically to the organs most in need. For example, a bio-engineered heme-containing protein, OMX-CV, has a 10-fold higher affinity for oxygen than hemoglobin does and has been shown to selectively deliver oxygen to hypoxic tissues but not to tissues at physiologic oxygen tension rates in juvenile lambs [121].

Investigations of the redox signaling pathways involved in oxidative stress have revealed other potential future targets for clinical study. Thioredoxin is an antioxidant enzyme that catalyzes the reduction of oxidized cysteine residues, playing an important role in reducing protein disulfide bonds. Alterations in thioredoxin and its partner enzyme, thioredoxin reductase, have been shown to affect susceptibility to hyperoxic lung injury [122]. Interestingly, the drugs auranofin and aurothioglucose inhibit thioredoxin reductase and have been shown to protect mice from hyperoxic lung injury through the induction of nuclear factor erythroid 2-related factor (NRF2)-induced genes [123,124]. NRF2 is an oxidant-activated transcription factor that stimulates the transcription of a variety of endogenous antioxidant systems [125]. Similarly, melatonin is a neurohormone that induces a variety of endogenous antioxidant enzymes and has been shown to mitigate hyperoxic lung injury in rats, as well as hypoxic ischemic encephalopathy in pigs [126,127,128]. In contrast to trials involving the direct application of antioxidant enzymes or cofactors, these medications provide an alternative approach through the induction of endogenous antioxidant systems.

Lastly, patient responses to hypoxia and hyperoxia may vary based on genetic differences between antioxidant systems. In a population of very low birth weight infants, sequencing of a variety of antioxidant genes revealed single-nucleotide polymorphisms in the NQO1 gene (coding for NADPH quinone reductase) and the NFE2L2 gene (coding for NRF2), which were associated significant differences in the risk of BPD [129]. As exome and genome sequencing become cheaper and more ubiquitous in the NICU, identification of genetic differences in endogenous antioxidant genes may allow for individualized approaches to mitigating oxidative injury.

## 5. Conclusions

Preterm and term newborns are unique populations with decreased antioxidant capacity and dependence on redox signaling for rapid growth and development. Despite advances in neonatal care, supplemental oxygen remains the most commonly used drug in the NICU. While oxygen can be life-saving, it is toxic in excess. Its application to neonatal medicine has seen a dramatic pendulum swing, from ubiquitous use both in the delivery room and beyond to intense restriction due to concern for retinal disease. The introduction of the pulse oximeter to initial newborn resuscitation and the NICU allowed for targeted oxygen therapy, while broad international collaboration has resulted in ample data to support current guidelines. Nonetheless, pulse oximetry provides limited information with which to make therapeutic decisions, and supplemental oxygen remains a blunt and untargeted therapy. Future strategies to optimize oxygen delivery while mitigating oxidative damage may involve NIRS monitoring or organ-, cell-, and patient-specific therapies that harness endogenous antioxidant systems.

## Data Availability

Not applicable.

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
