# Peer review of "Supplemental Oxygen in the Newborn: Historical Perspective and Current Trends"

_antioxidants, 2021, doi:10.3390/antiox10121879_

Round 1
Reviewer 1 Report
This is a very well-written mms that provides up to date information regarding the role and the problems associated with oxygen supplementation in the neonate. Although many reviews have been published related to O2 use in the adult, there are special and serious issues related to this therapy in the neonate, as is well known with respect to role of eccess O2 in newborn retinal disease. This review adequately covers all ofd the major issues and studies. The following comments are only for minor issues.
Minor comments:
1) line 64-- what does mitochondrial oxidative potential mean in this context?
2) line 157-- add 'in'.
3) line 179--babies?
4) line 209 and 236--is post-menstrual a common way to describe the age of a baby?
5) Figure 1-- I see limited justification for including this figure in the mms-- adds nothing. Might be a little more interesting if it showed HbF.
6) line 355--add 'with'.
7) line 367-- is 'targeted' the right word here?
Reviewer 2 Report
Mathias et al. submitted a review detailing the history of oxygen therapy in newborns, with particular focus on the pathophysiology of hyperoxic injury.
The paper is generally well written and richly referenced with respect to both the basic science and clinical evidence of oxygen therapy.
One section that seems unnecessary is the description of the principles underlying the function of pulse oximetry, in lines 248-258, as well as in Figure 1. The information contained therein is very basic, and it does not provide fundamental support to other aspects of the manuscript, nor is it presented in a manner that provides some new insight.
Conversely, there are 2 common areas of neonatal oxygen therapy that could benefit from some discussion by the authors. First, at the end of the paragraph ending in line 244, or is a new paragraph, the authors may consider emphasizing the notion that optimal oxygen targets may vary over time, to which the previously alluded in passing. They may refer, for example, to the results from the STOP-ROP trial, whey they already include as reference 75.
Another common use of oxygen in term and near-term neonates that is not addressed at all but could benefit from a brief discussion by these authors is the use pulmonary hyperoxia (e.g., prolonged exposure to FiO2 of 80% to 100%), even with concurrent systemic hyperoxia, with the purpose of achieving maximal pulmonary vasodilation in patients with persistent pulmonary hypertension. There is evidence of widely variable clinical practice in this regard. A basic discussion of the laboratory and clinical evidence supporting the therapeutic use of hyperoxic pulmonary vasodilatation in PPHN, as well as its potential adverse effects on pulmonary vasodilatory mechanisms and other aspects of lung function, along with a couple of supporting references, would add to the value of this paper.
